# Association of Nutritional Status and Diet Diversity with Skeletal Muscle Strength and Quality of Life among Older Arab Adults: A Cross-Sectional Study

**DOI:** 10.3390/nu15204382

**Published:** 2023-10-16

**Authors:** Rahaf Alotaibi, Alanoud Aladel, Sulaiman A. Alshammari, Mahmoud M. A. Abulmeaty, Adel A. Alhamdan

**Affiliations:** 1Department of Community Health Sciences, College of Applied Medical Sciences, King Saud University, Riyadh 11451, Saudi Arabia; rahaf.abdullhadi@gmail.com (R.A.); aaladel@ksu.edu.sa (A.A.); mabulmeaty@ksu.edu.sa (M.M.A.A.); 2Department of Family and Community Medicine, College of Medicine, King Saud University, Riyadh 11461, Saudi Arabia; sulaiman@ksu.edu.sa

**Keywords:** older adults, nutritional status, skeletal muscle strength

## Abstract

There is little research evaluating skeletal muscle strength, nutritional status, and quality of life in older Arab adults. This study examined the association of nutritional status with skeletal muscle strength and quality of life among older adults living in Saudi Arabia. A cross-sectional study was conducted among older adults (*n* = 166 older adults; 57.8% females) who visited primary outpatient clinics at King Khalid University Hospital in Riyadh City. Sociodemographic data, Mini Nutritional Assessment short-form (MNA-SF), diet variety score (DVS), and health-related quality of life (HR-QoL) were assessed. Furthermore, handgrip strength (HGS) and knee extension strength (KES) were measured to evaluate skeletal muscle strength. Malnourishment and risk of malnutrition were found in 16.9% of our sample. Nutritional status was significantly associated with muscle strength and HR-QoL (*p* < 0.05). The well-nourished group had higher HGS, KES, and total HR-QoL scores compared to the at risk of malnutrition or malnourished group. Moreover, there was a moderate positive correlation between MNA-SF and total HR-QoL scores (r = 0.40). The percentage of individuals in the well-nourished group categorized with a high or moderate DVS was significantly higher than those at risk of malnutrition or are malnourished; however, DVS was not associated with muscle strength and HR-QoL. The MNA-SF score had a significant positive correlation with HGS (r = 0.30) and KES (r = 0.23). An increase in the MNA-SF score was significantly associated with higher odds of being moderate/high (HGS and HR-QoL) in the crude and adjusted models. In conclusion, maintaining adequate nutritional status is beneficial for preserving skeletal muscle strength and promoting better HR-QoL among older individuals. Therefore, applications of appropriate nutritional and muscle strength assessments in geriatric care institutions are recommended.

## 1. Introduction

The global population aged ≥60 is projected to increase from 1 billion in 2020 to 1.4 billion by 2030; by 2050, it is expected to double, reaching 2.1 billion [1]. In the Kingdom of Saudi Arabia (KSA), older adults aged ≥60 years are anticipated to increase fivefold, from 2 million to 10.5 million between 2020 and 2050 [2]. The dramatic change in the population structure will increase the risk of disabilities and chronic diseases in older adults, both globally and in the KSA [3,4].

The aging process may be associated with various conditions, such as malnutrition, sensory impairments, physical injury, decreased cognitive and mental health, reduced muscle strength, and decreased health-related quality of life (HR-QoL) [5,6]. Regarding HR-QoL, the Short Form-36 (SF-36) questionnaire is considered the most common, widespread, reliable, and validated tool to assess the quality of life in older adults [7,8]. Most older adults can be anticipated to have nutritional and health issues that will adversely affect their HR-QoL and ability to perform daily tasks independently [9]. High numbers of older adults have malnutrition, which could be diagnosed by healthcare providers using various instruments, such as the Mini Nutritional Assessment (MNA) questionnaire, the Mini Nutritional Assessment short-form (MNA-SF), and the diet variety score (DVS) [10,11]. In this regard, several studies have claimed the validity and reliability of the MNA and MNA-SF questionnaires for determining malnutrition and measuring nutritional status in older adults [12,13]. In addition, dietary diversity has been considered a valid and essential component of a high-quality diet, and it is a simple count of food items or food groups consumed by an individual over a certain period [14]. One of the most common health issues in older adults is decreased muscle strength [15]. Currently, easy and valid methods to determine skeletal muscle strength are handgrip strength (HGS) and knee extension strength (KES) tests [16,17]. Measuring skeletal muscle strength indicates the physical and functional status of older adults [17]. HGS and KES are associated with psychological, functional, and physical performance [17]. There is little research on the association between nutritional status, using both the MNA-SF and diet diversity score, and muscle strength as well as quality of life in older adults. It has not been studied in the Kingdom of Saudi Arabia (KSA). Taken together, further investigation into the associations between nutritional status, diet diversity, and critical health outcomes in older adults is required [18]. Thus, this study aims to investigate whether nutritional status and diet diversity are associated with skeletal muscle strength and HR-QoL in older adults. 

## 2. Materials and Methods

### 2.1. Study Design and Sample Selection

This study included older adults (≥60 years old) who visited the primary outpatient clinics at King Khalid University Hospital in Riyadh, the KSA, between January and March 2023. Individuals with severe cognitive impairments or those unable to provide informed consent were excluded. Furthermore, the exclusion criteria included the following: cancer, neurological, or major musculoskeletal disorders and deformation of body parts. In addition, patients who were hospitalized, underwent surgery within the previous six months, received nutritional supplements, and had missing data were excluded. The sample size was estimated using G*Power software (version 3.1.9.7) based on the point biserial correlation test and an effect size of 0.2531, according to the correlation between nutritional status and HGS [19], 5% significance level, and 80% power. The minimum sample size was estimated at 117 participants. We assumed a response rate of 90%, resulting in a total sample size of 130. Ethical approval was obtained from the College of Medicine, the Institutional Review Board (IRB) Ethics Committee at King Saud University (No. 23/0049/IRB; approval date: 23 January 2023). 

### 2.2. Sociodemographic Data and Anthropometric Measurements 

Data were collected after IRB approval via face-to-face interviews. Sociodemographic data were taken, including age, gender, income level, educational level, marital status, medication, history of health conditions, and number of comorbidities. A weight scale (Seca Co., Hamburg, Germany) was used to measure weight and height. Afterward, BMI was calculated as kg/m^2^. Calf circumference (CC) was measured in cm using a standard measuring tape. 

### 2.3. Skeletal Muscle Strength

Skeletal muscle strength was assessed using HGS and KES. Both measurements are considered the most common ways to determine muscle strength in the epidemiological and clinical practice arena, and they are reliable and valid measures in older adults [20,21,22].

#### 2.3.1. Handgrip Strength (HGS) 

For HGS, a Jamar Hydraulic Dynamometer (Jamar, model 5030 J1, Sammons Preston Rolyan, Bolingbrook, IL, USA) was used to assess the muscle strength of the dominant hand. The participants stood upright with their elbows fully extended and squeezed the grip continuously at maximum force for at least 3 s. We used the maximal HGS among the recorded values for statistical analyses [23]. Participant HGS was measured twice to reduce measurement errors, and the highest value obtained was recorded. 

#### 2.3.2. Knee Extension Strength (KES)

KES was used to assess the dominant leg by using a hand-held dynamometer (HHD) manual muscle tester (Lafayette Manual Muscle Tester Model 01165, Lafayette Instrument Company, Lafayette, IN, USA). The participants were seated with arms folded across their chest, and their hip and knee joint angles were set at 90 degrees of flexion, with their feet elevated above the floor. The participant applied maximal force to the HHD for three seconds, which was held stationary by the investigator at about 10 cm above the ankle joint [24]. The mean peak force from three trials was calculated. 

### 2.4. Nutritional Screening and Assessment 

#### 2.4.1. Mini Nutritional Assessment Short-Form (MNA-SF) 

The MNA-SF is a reliable, validated tool that has been widely used to assess the nutritional status of older adults [25,26,27]. The MNA-SF scale (0–14 points) consists of six weighted questions. Depending on the test score, older adults were classified into the following categories: well-nourished (12–14 MNA points), at risk of malnutrition (8–11 MNA points), or malnourished (MNA < 8 points) [27]. 

#### 2.4.2. Dietary Variety Score (DVS)

The dietary variety score (DVS) was used to estimate the diet diversity of older adults, and it refers to the number of different food groups consumed by an individual over a certain period [9]. The consumption frequency of the five main food groups (meat and eggs, dairy products, fruits, vegetables, and bread/grains/cereals) was determined based on weekly consumption. To qualify as a “food-group consumer”, a participant should eat at least one serving of a specific food group daily. A food group consumed every day received a score of one, whereas a food group that was not consumed every day received a score of zero.

Furthermore, the food frequency score (FFS) was used to assess diet variety based on a one-week food frequency questionnaire, and it was classified according to the frequency of consumption of each food group [28,29]. A score ranging from 0 to 3 points was assigned to indicate the frequency of consumption for each food group: consumed almost every day (3 points), consumed 3–4 days per week (2 points), consumed 1–2 days per week (1 point), and rarely consumed (0 points). Consequently, the FFS was determined by summing the scores for each of the five food groups (range, 0–15 points) and then categorized into three tertile cutoff points (high DVS, moderate DVS, or low DVS). DVS and FFS measurements were estimated based on the validated Saudi food frequency questionnaire to reflect the most consumed food items and dietary habits of the Saudi community [30]. 

### 2.5. Health-Related Quality of Life (HR-QoL)

HR-QoL was assessed using the Short Form-36 (SF-36) questionnaire, which is a globally used tool for assessing HR-QoL and has been validated in different populations [31]. HR-QoL domains include physical functioning, physical health, painless body, general health, total physical components, emotional health, energetic body, emotional well-being, social functioning, total mental components, and total HR-QoL score [31]. The Arabic version of the SF-36, which has also been validated in older adults [32], was used in this study. 

### 2.6. Statistical Analysis

The data were entered and analyzed using Statistical Package for Social Sciences (SPSS) version 22. The Shapiro–Wilk test was used to test data normality. Continuous variables were displayed as median and interquartile range. For HR-QoL, the outcome of the variables was presented as mean ± SD of percentages. Categorical variables were presented as frequencies and percentages. For categorical outcomes, comparisons between groups were performed using the chi-square test or Fisher’s exact test, as appropriate. For comparison of the continuous outcomes, the Mann–Whitney test and Kruskal–Wallis test were used, as appropriate. Correlations between continuous variables were examined using Spearman’s correlation coefficient. Spearman’s rank correlation coefficient was interpreted as zero (=0), weak (≥0.1 and <0.4), moderate (≥0.4 and <0.7), strong (≥0.7 and <1), and perfect (=1) [33]. In addition, binary logistic regression was utilized to estimate the odds ratio and assess the factors associated with nutritional status. HGS, KES, and HR-QoL values were divided into three tertiles (high, moderate, or low); then, the high and moderate tertiles were combined as one group. Individuals with HGS equal to or less than 19 were considered in the low tertile, those between 19 and 23 were in the moderate tertile, and those between 23.1 and 43 were considered in the high tertile. For KES, individuals with values ≤ 9.5 were considered in the low tertile, the moderate tertile was considered for those with values between 9.5 and 11.82, and the high tertile was considered for those with values between 11.82 and 23.00. HR-QoL tertiles were designed similarly, i.e., low < 59.86; moderate, 59.86 to 82.00; and high, 82.00 to 112.78. Binary logistic regressions were presented in three models: model 1—the crude model; model 2—adjusted by total DVS for the independent variable MNA-SF and adjusted by MNA-SF score for the independent variable total DVS; and model 3—adjusted by age, gender, disease, and variables in model 2. A two-sided *p*-value ≤ 0.05 was considered statistically significant.

## 3. Results

The total number of initially identified participants was *n* = 778. Participants excluded before screening were based on two criteria: older adults who refused to participate in the study (*n* = 420) and people less than 60 years old (*n* = 114). A total of 244 older adults were screened. After the screening, 31 participants who refused to complete the study, 4 participants who had surgery within the previous six months, and 12 participants who had pain in their extremities were excluded. In addition, 31 participants were excluded from the analysis due to missing data. Thus, 166 participants were finally enrolled in the present study, as shown in the STROBE flow chart (Figure 1).

### 3.1. General Characteristics and Nutritional Status of Older Adults

Table 1 shows the 166 participants (42.2% male and 57.8% female) included in the study, with an age range between 60 and 84 years (median, 66 years; IQR, 8 years). The majority of participants were married (68.1%). Those who were illiterate and with a monthly income of less than 3000 SR comprised 38% of the total study sample. Most participants had at least one disease (97.6%), with diabetes mellitus (DM) and hypertension (HTN) being the main comorbidities found in 71.7% and 69.3% of older adults, respectively. About 95.8% of the study sample were using medication, and 28.3% had two diseases. According to the MNA-SF, 83.1% of the participants were considered well-nourished, while 16.9% were at risk of malnutrition or malnourished. Due to the small percentage of participants being classified as malnourished (0.6%), they were combined with those classified as at risk of malnutrition as one group, named the at risk of malnutrition or malnourished group. Interestingly, a significant relationship was found between nutritional status and academic level (*p =* 0.010), indicating that illiterate participants were more likely to be at risk of malnutrition or show evident malnutrition. Additionally, osteoporosis was significantly associated with a risk of malnutrition or malnourishment in older adults (*p* = 0.001).

### 3.2. Muscle Strength, Calf Circumference, and Body Mass Index

Table 2 describes muscle strength, CC, and BMI according to the nutritional status among older adults. The results show that the at risk of malnutrition or malnourished group had significantly lower HGS and KES than the well-nourished group (*p* < 0.05). However, there were no significant differences in CC and BMI between the two groups. Table 3 shows the correlation among MNA-SF, muscle strength, BMI, and CC. The results exhibit a significant positive correlation between MNA-SF and HGS in all participants (r = 0.30). In addition, a significant positive correlation was found between MNA-SF and KES (r = 0.23). The correlation between HGS and KES was significantly positive in the total study sample (r = 0.54). In addition, a significant positive correlation between BMI and CC (r = 0.70) was observed.

### 3.3. Health-Related Quality of Life and Nutritional Status

The HR-QoL of older adults according to their nutritional status showed that participants categorized in the at risk of malnutrition or malnourished group had significantly lower scores of total HR-QoL (56.21 ± 14.29 vs. 73.70 ± 17.35; *p* = 0.001). Additionally, physical and mental components were significantly higher in the well-nourished group compared to the at risk of malnutrition or malnourished group, as shown in Figure 2. There was a positive correlation between the MNA-SF score and all components of HR-QoL (Table 4). The correlation was statistically significantly associated with total physical components (r = 0.35), total mental components (r = 0.33), and total HR-QoL scores (r = 0.40), indicating a moderate correlation with the MNA-SF. 

### 3.4. Diet Variety Score (DVS)

Table 5 displays DVS according to nutritional status. There was no significant difference in the daily consumption of each food group and total food groups between the well-nourished and the at risk of malnutrition or malnourished group. However, when DVS was classified into low, moderate, or high categories, the results showed that the percentage of high or moderate DVS categories of the well-nourished group was higher than the at risk of malnutrition or malnourished group (*p =* 0.037). DVS showed no significant differences in muscle strength, CC, BMI, and HR-QoL variables concerning DVS categories (low, moderate, or high). In addition, there was no significant correlation between diet variety and muscle strength, BMI, or CC. However, the correlations between DVS and HR-QoL indicate that DVS had a significant and weak correlation with total physical components (*r =* 0.19) and total HR-QoL score (*r =* 0.16), as presented in Table 6.

### 3.5. Binary Logistic Regression According to Nutritional Status of the Study Participants 

Table 7 presents the binary logistic regression for HGS using the moderate/high HGS group as the reference category, according to the MNA score and total DVS. The findings indicate that the increase in the MNA-SF score was significantly associated with higher odds of exhibiting moderate or high HGS in model 1 (OR = 1.33; CI: 1.08–1.64), model 2 (OR = 1.32; CI: 1.07–1.63), and model 3 (OR = 1.30; CI: 1.01–1.72). However, the total DVS was not significantly associated with HGS in all three models. The binary logistic regression for KES using the moderate/high KES as the reference category according to the MNA score and total DVS is presented in Table 8. In model 1, the increase in the MNA-SF score was significantly associated with an increase in the odds of having moderate or high KES (OR = 1.19; CI: 1.10–1.44). However, there was no significant association between the MNA-SF score and KES in models 2 and 3. Additionally, the total DVS was not significantly associated with KES in all three models. The binary logistic regression for HR-QoL, using the moderate/high HR-QoL group, according to the MNA and total DVS, is shown in Table 9. In all three models, the increase in the MNA-SF score was significantly associated with an increase in the odds of having moderate or high HR-QoL in model 1 (OR = 1.62; CI: 1.30–2.03), model 2 (OR = 1.64; CI: 1.30–2.07), and model 3 (OR = 1.64; CI: 1.23–2.18). Additionally, the total DVS was significant in model 1 (OR = 1.24; CI: 1.05–1.47), while there was no significant association in models 2 and 3.

## 4. Discussion

The present study investigated the associations between nutritional status, skeletal muscle strength, and HR-QoL among older adults. Based on the MNA-SF, the percentage of individuals at risk of malnutrition or malnourished was 16.9% among older Saudi adults (malnourished, 0.6%; at risk of malnutrition, 16.3%). However, numerous published works have shown that the prevalence of older adults at risk of malnutrition or malnourished varies. For example, according to the MNA-SF, a Turkish study estimated the prevalence of malnutrition at 4.2% and the risk of malnutrition at 21.9% [34]. Previous studies conducted in China using the MNA-SF found malnutrition ranged from 10.3% to 32.4% among older adults [35,36]. A cross-sectional study conducted in Finland reported the prevalence of malnutrition at 18% and the risk of malnutrition at 64% [37]. In addition, Cereda et al. conducted a meta-analysis of 240 studies and reported that the prevalence of malnutrition varied between 14% and 21% [38]. Malnutrition among older adults is considered a global issue, and applying nutrition screening protocols becomes necessary to detect malnutrition at an earlier stage [36,39,40].

Based on the socio-demographic and clinical characteristics of the study participants, those who were illiterate were more likely at risk of malnutrition (or malnourished) than well-nourished participants. A previous study identified a significant association between nutritional status, marital status, and educational level [41]. In contrast, a study conducted by Hua et al. found no significant relationship between MNA-SF and education level. However, consistent with our results, a Chinese study identified no association between marital and nutritional status among older adults [36]. Moreover, in our study, osteoporosis was significantly related to nutritional status, with a higher number of osteoporotic older adults categorized in the at risk of malnutrition or malnourished group than the well-nourished group. Likewise, a study conducted by Zhang et al. showed that the prevalence of osteoporosis was markedly higher in malnourished older adults compared with well-nourished older adults [35].

HGS and KES were assessed to determine muscle strength in the upper and lower muscles, respectively. Several studies, including this study, found that nutritional status was significantly associated with skeletal muscle strength [42,43]. In contrast, a study conducted by Hua et al. revealed no significant association between the MNA-SF and skeletal muscle strength measured using a digital hand dynamometer [36]. In addition, Hua’s study [36] found a significant association between nutritional status and BMI and CC, while no significant association was found in the present study. An Indonesian study of 98 older adults from geriatric outpatient clinics showed a significant weak correlation (r = 0.27) between nutritional status and HGS [43], while our study found a moderate correlation (r = 0.30). According to Debia et al., the MNA-SF score indicated a moderate positive correlation with BMI and HGS and a strong correlation with CC among older Brazilian adults, and they suggested that the higher the BMI, HGS, and CC, the better the nutritional status, strength, and functionality [42]. This could be because body composition changes in older adults as they age, and some studies have suggested that a slightly higher BMI may be related to improved health outcomes and longevity in older adults compared to younger adults [44,45,46]. 

Regarding HGS, our study found that the MNA-SF was significantly correlated with muscle strength (HGS and KES) in the total sample. A study conducted in the Netherlands assessed dietary parameters based on the short nutritional assessment questionnaire among 299 older adults, specifically focusing on KES. The study found that dietary, physical, and psychological factors exhibited a stronger association with KES than HGS among 163 older adults [17]. In contrast, our study found that HGS is moderately associated with nutritional status and weakly associated with KES. Furthermore, KES and HGS may be valuable indicators for functional performance screening among older adults [17]. 

Several factors might impact muscle strength, including gender differences, age, grip size, dominance, anthropometric variables, genetic factors, muscle fiber, body composition, and training experience [47,48,49,50,51]. The regression analysis in this study showed that the MNA-SF, rather than the DVS, was a better predictor of muscle strength, as denoted by HGS and KES. A recent study reported a significant association between the MNA-SF and evident sarcopenia in a nourished population. However, the MNA-SF score showed significant differences in muscle strength, performance, and the incidence of sarcopenia, even within the nourished group [52].

With regard to diet variety, a previous study of older adults at risk of malnutrition or malnourished reported a significant positive association with moderate/inadequate DVS (RR = 2.04) and a significant negative association with protein (RR = 0.76) [36]. In addition, a high DVS was significantly associated with being well nourished, similar to our study [36]. A previous study found that older adults with low HGS had a low intake of specific nutrients, such as proteins, essential amino acids, calcium, vitamin D, and antioxidants [53]. Nevertheless, our study did not evaluate micronutrients or find a significant correlation in each food group. In the present study, we found a significant association between HGS and nutritional status classified as well-nourished or at risk of malnutrition/malnourished using the MNA-SF. On the other hand, we did not find an association between HGS and DVS classified into low, moderate, or high. Similar to our finding, a Japanese study found no significant relationship between KES and DVS in older women [54]. Concerning HGS, a Japanese prospective study examined the association of dietary variety with changes in lean mass, HGS, and usual gait speed over four years in older adults; it revealed that high dietary variety preserves HGS and usual gait speed but not lean mass [55]. However, there are several possible reasons why there may not be a significant relationship between DVS and muscle strength. In addition, dietary diversity, muscle strength factors include physical activity, age, heredity, and medical conditions [51]. Furthermore, older adults may have difficulty recalling their nutrition intake, and it is preferable to involve caregivers in recalling nutritional surveys using detailed food frequency questionnaires when assessing DVS.

Likewise, a Spanish longitudinal study found no significant association between Mediterranean diet scores as an indication of DVS and HR-QoL in physical and mental components among older adults [56]. In contrast, an Australian study found that DVS, assessed by adherence to the Australian dietary guidelines, was prospectively associated with a significantly better quality of life after five years [57]. These discrepancies in the above studies could be related to the methods of measuring DVS. In the present study, the lack of a detailed quantitative nutrition evaluation may have contributed to non-significant associations. Optimal nutrition improves HR-QoL in older adults by promoting health and preventing dietary insufficiency [58]. Thus, assessing nutritional status followed by appropriate diet intervention should be prioritized to improve HR-QoL in older people. Older adults with malnutrition/malnutrition risk based on MNA and MNA-SF have been shown to have a lower HR-QoL score across all domains (*p* < 0.05) [34]. A meta-analysis reported that older adults with malnutrition are more likely to have poor HR-QoL (overall OR: 2.85), and good nutritional status can cause significant improvements in HR-QoL, both in physical and mental aspects [59]. This is consistent with our results, in which HR-QoL variables were significantly lower in older adults classified as at risk of malnutrition or malnourished compared to well-nourished older adults, except in emotional well-being. Khatami et al. showed a significant correlation between MNA and HR-QoL across all domains among older adults using the same questionnaire as our study group (SF-36). In addition, they found that those categorized as malnourished or at risk of being malnourished had considerably inferior HR-QoL compared to well-nourished subjects [41]. This could suggest that nutritional status may be an essential factor that affects HR-QoL. 

To the best of our knowledge, this is the first Saudi study investigating the association between nutritional status using MNA-SF, diet diversity, muscle strength, and quality of life in older adults. Furthermore, HGS was used, which is considered a potentially rapid, simple, noninvasive, objective, and highly reliable tool for determining nutritional status [19,20]. In addition, face-to-face interviews were conducted to collect subjective data instead of using a written survey; this may improve the response rate, provide more detailed data, and allow illiterate older adults to engage in the study. 

This study includes several limitations. Firstly, this study cannot establish a causal association because it is cross-sectional. Secondly, the study was conducted at a single center in Riyadh City, which might limit the generalization of the results. Thirdly, we did not measure muscle mass and nutritional biomarkers in the blood. Finally, a detailed physical activity survey was not included in our study, which may impact the results for muscle strength. Comprehensive cohort and intervention studies investigating the cause-and-effect relationship between nutritional status and skeletal muscle strength, as well as HR-QoL, are recommended.

## 5. Conclusions

In conclusion, this study found a significant association between nutritional status, skeletal muscle strength, and HR-QoL in older adults. Furthermore, those categorized as well-nourished had higher HGS, KES, and HR-QoL scores than those categorized as at risk of malnutrition or malnourished. This may indicate that optimal nutrition is crucial for older adults to maintain their muscle strength and improve their quality of life as they age. Therefore, healthcare professionals should prioritize nutritional assessment, including measuring HGS and KES, and counseling as a part of routine care for the older adult population. By ensuring optimal nutritional status for older adults, we can assist them in maintaining their independence and wellness as they age. 

## Figures and Tables

**Figure 1 nutrients-15-04382-f001:**
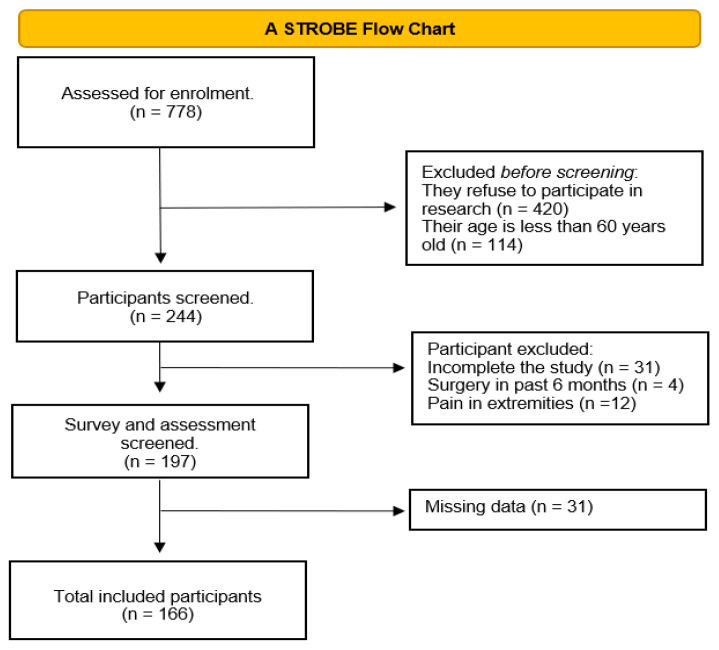
The STROBE flow chart of the study.

**Figure 2 nutrients-15-04382-f002:**
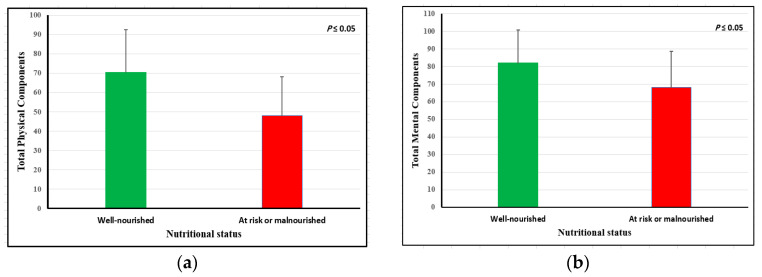
(**a**) Total physical components and (**b**) total mental components.

**Table 1 nutrients-15-04382-t001:** General characteristics of study participants according to their nutritional status.

	MNA-SF			
Variable	All *n* = 166	Well-Nourished *n* = 138	At Risk of Malnutrition or Malnourished *n* = 28	*p*-Value
Age	66 (8)	66 (8)	67 (9)	0.416
Gender				0.440
Male	70 (42.2%)	63 (45.7%)	7 (25%)	
Female	96 (57.8%)	75 (54.3%)	21(75%)	
Academic level				0.010
Illiterate	63 (38.0%)	48 (34.8%)	15 (53.6%)	
School	53 (31.9%)	41 (29.7%)	12 (42.9%)	
Diploma or bachelor	46 (27.7%)	45 (32.6%)	1 (3.6%)	
Postgraduate	4 (2.4%)	4 (2.9%)	0 (0.0%)	
Marital status				0.328
Single	3 (1.8%)	2 (1.4%)	1 (3.6%)	
Married	113 (68.1%)	98 (71.0%)	15 (53.6%)	
Divorced	13 (7.8%)	10 (7.2%)	3 (10.7%)	
Widowed	37 (22.3%)	28 (20.3%)	9 (32.1%)	
Monthly income				0.220
<3000 SR	63 (38.0%)	51 (37.0%)	12 (42.9%)	
3000–5999 SR	37 (22.3%)	26 (18.8%)	11(39.3%)	
6000–10,000 SR	21 (12.7%)	18 (13.0%)	3 (10.7%)	
>10,000 SR	45 (27.1%)	43 (31.2%)	2 (7.1%)	
Older adults have a disease	161 (97.0%)	133 (96.4%)	28 (100%)	0.590
Older adults on medication	159 (95.8%)	132 (95.7%)	27 (96.4%)	1.000
Comorbidities				
Hypercholesterolemia	43 (25.9%)	34 (24.6%)	9 (32.1%)	0.409
CVD	17 (10.2%)	12 (8.7%)	5 (17.9%)	0.170
HTN	115 (69.3%)	93 (67.4%)	22 (78.6%)	0.242
Osteoporosis	43 (25.9%)	29 (21%)	14 (50%)	0.001
DM	119 (71.7%)	98 (71%)	21 (75%)	0.670
Other	75 (45.2%)	59 (42.8%)	16 (57.1%)	0.163
Number of comorbidities				0.071
None	4 (2.4%)	4 (2.9%)	0 (0.0%)	
One	36 (21.7%)	34 (24.6%)	2 (7.1%)	
Two	47 (28.3%)	40 (29%)	7 (25%)	
Three	41 (24.7%)	34 (24.6%)	7 (25%)	
Four	31 (18.7%)	21(15.2%)	10 (35.7%)	
Five or more	7 (4.2%)	5 (3.6%)	2 (7.1%)	

MNA-SF—Mini Nutritional Assessment short-form; HTN—hypertension; CVD—cardiovascular disease; DM—diabetes mellitus. For age, data are presented as median and IQR. Other variables are presented as frequencies and percentages. Significance at *p*-value < 0.05.

**Table 2 nutrients-15-04382-t002:** Muscle strength, calf circumference, and body mass index of study participants according to their nutritional status.

	MNA-SF			
Variable	All *n* = 166	Well-Nourished *n* = 138	At Risk of Malnutrition or Malnourished *n* = 28	*p*-Value
HGS (kg)	20 (14)	21 (12.25)	17.5 (9.75)	0.001
KES (kg)	10.85 (3.7)	11 (3.70)	10.1 (4.03)	0.048
CC (cm)	36 (5.8)	36 (5.13)	36.5 (11.75)	0.943
BMI (kg/m^2^)	29.3 (7.74)	29.6 (7.35)	28.38 (13.94)	0.380

MNA-SF—Mini Nutritional Assessment short-form; HGS—handgrip strength; KES—knee extension strength; CC—calf circumference; BMI—body mass index. Data are presented as median and IQR. Significance at *p*-value < 0.05.

**Table 3 nutrients-15-04382-t003:** Correlations between muscle strength, calf circumference, body mass index, and Mini Nutritional Assessment short-form scores.

Variable	HGS (Kg)	KES (Kg)	BMI (kg/m^2^)	CC (CM)
All				
MNA-SF scores	0.30 **	0.23 **	0.080	0.09
HGS (Kg)	-	0.54 **	−0.10	−0.02
KES (Kg)	-	-	−0.06	−0.12
BMI (kg/m^2^)	-	-	-	0.70 **

MNA-SF—Mini Nutritional Assessment short-form; HGS—handgrip strength; KES—knee extension strength; CC—calf circumference; BMI—body mass index. ** Spearman’s rho correlation is significant at the 0.01 level.

**Table 4 nutrients-15-04382-t004:** The correlation between Mini Nutritional Assessment short-form and health-related quality of life.

	MNA-SF Score	
Variable	r	*p* Value
Total physical components	0.35	0.001
Total mental components	0.33	0.001
Total HR-QoL score	0.40	0.001

MNA-SF—Mini Nutritional Assessment short-form; HR-QoL—health-related quality of life. Spearman’s rho correlation is significant at *p*-value < 0.05.

**Table 5 nutrients-15-04382-t005:** Diet variety according to the nutritional status of the study participants.

	DVS			
Variable	All *n* = 166	Well-Nourished *n* = 138	At Risk of Malnutrition or Malnourished *n* = 28	*p*-Value
Daily intake of food groups				
Daily intake of dairy products	69 (41.6%)	59 (42.8%)	10 (35.7%)	0.491
Daily intake of fruits	136 (81.9%)	116 (84.1%)	20 (71.4%)	0.113
Daily intake of vegetables	96 (57.8%)	76 (55.1%)	20 (71.4%)	0.110
Daily intake of meats and eggs	95 (57.2%)	79 (57.2%)	16 (57.1%)	0.992
Daily intake of carbohydrates	136 (81.9%)	113 (81.9%)	23 (82.1%)	0.974
Daily intake of all food groups	23 (13.9%)	21 (15.2%)	2 (7.1%)	0.373
Dairy products				0.223
Rarely	16 (9.6%)	11 (8.0%)	5 (17.9%)	
1–2 times/week	34 (20.5%)	29 (21%)	5 (17.9%)	
3–4 times/week	30 (18.1%)	23(16.7%)	7 (25.0%)	
5–7 times/week	86 (51.8%)	75 (54.3%)	11 (39.3%)	
Fruits				0.247
Rarely	1 (0.6%)	1 (0.7%)	0 (0.0%)	
1–2 times/week	5 (3.0%)	3 (2.2%)	2 (7.1%)	
3–4 times/week	9 (5.4%)	6 (4.3%)	3 (10.7%)	
5–7 times/week	151 (91.0%)	128 (92.8%)	23 (82.1%)	
Vegetables				0.343
Rarely	3 (1.8%)	2 (1.4%)	1 (3.6%)	
1–2 times/week	23 (13.9%)	18 (13.0%)	5 (17.9%)	
3–4 times/week	30 (18.1%)	28 (20.3%)	2 (7.1%)	
5–7 times/week	110 (66.3%)	90 (65.2%)	20 (71.4%)	
Meats and eggs				0.562
Rarely	1 (0.6%)	1 (0.7%)	0 (0.0%)	
1–2 times/week	19 (11.4%)	14 (10.1%)	5 (17.9%)	
3–4 times/week	27 (16.3%)	24 (17.4%)	3 (10.7%)	
5–7 times/week	119 (71.7%)	99 (71.7%)	20 (71.4%)	
Breads/grains/cereals				0.712
Rarely	2 (1.2%)	2 (1.4%)	0 (0.0%)	
1–2 times/week	8 (4.8%)	7 (5.1%)	1 (3.6%)	
3–4 times/week	11 (6.6%)	8 (5.8%)	3 (10.7%)	
5–7 times/week	145 (87.3%)	121(87.7%)	24 (85.7%)	
DVS categories				0.037
Low	60 (36.1%)	44 (31.9%)	16 (57.1%)	
Moderate	64 (38.6%)	56 (40.6%)	8 (28.6%)	
High	42 (25.3%)	38 (27.5%)	4 (14.3%)	

MNA-SF—Mini Nutritional Assessment short-form; DVS—diet variety score. Data are presented as number (*n*) and percentage (%). Significance at *p*-value ≤ 0.05.

**Table 6 nutrients-15-04382-t006:** Correlation between diet variety score and health-related quality of life.

	DVS	
Variable	r	*p*-Value
Total physical components	0.19 *	0.016
Total mental components	0.09	0.253
Total HR-QoL score	0.16 *	0.039

DVS—diet variety score; HR-QoL—health-related quality of life. * Spearman’s rho correlation is significant at *p*-value < 0.05.

**Table 7 nutrients-15-04382-t007:** Binary logistic regression for handgrip strength according to the MNA score and total DVS.

Variable	Model 1 OR (95% CI)	Model 2 OR (95% CI)	Model 3 OR (95% CI)
Total MNA-SF score	1.33 (1.08–1.64) *	1.32 (1.07–1.63) *	1.30 (1.01–1.72) *
Total DVS	1.07 (0.91–1.26)	1.11 (0.88–1.41)	1.13 (0.83–1.53)

Moderate or high HGS was used as the reference category. MNA-SF—Mini Nutritional Assessment short-form; HGS—handgrip strength; and DVS—diet variety score. Model 1—crude; model 2—adjusted by variables in the table (total MNA-SF scores and total DVS); model 3—adjusted by gender, age, diseases, and variables in the table (total MNA-SF scores and total DVS). Data are presented as an odds ratio (OR) and confidence interval (CI). * Significant at *p*-value ≤ 0.05.

**Table 8 nutrients-15-04382-t008:** Binary logistic regression for knee extension strength according to the MNA score and total DVS.

Variable	Model 1 OR (95% CI)	Model 2 OR (95% CI)	Model 3 OR (95% CI)
Total MNA-SF score	1.19 (1.10–1.44) *	1.18 (0.96–1.45)	1.09 (0.86–1.38)
Total DVS	1.12(0.95–1.32)	1.04 (0.82–1.32)	1.02 (0.77–1.33)

Moderate or high KES was used as reference categories. MNA-SF—Mini Nutritional Assessment short-form; KES—knee extension strength; and DVS—diet variety score. Model 1—crude; model 2—adjusted by variables in the table (total MNA-SF scores and total DVS); model 3—adjusted by gender, age, diseases, and variables in the table (total MNA-SF scores and total DVS). Data are presented as odds ratio (OR) and confidence interval (CI). * Significant at *p*-value ≤ 0.05.

**Table 9 nutrients-15-04382-t009:** Binary logistic regression for health-related quality of life according to the MNA score and total DVS.

Variable	Model 1 OR (95% CI)	Model 2 OR (95% CI)	Model 3 OR (95% CI)
Total MNA-SF score	1.62 (1.30–2.03) *	1.64 (1.30–2.07) *	1.64 (1.23–2.18) *
Total DVS	1.24 (1.05–1.47) *	1.12 (0.87–1.44)	1.25 (0.91–1.71)

Moderate or high HR-QoL was used as the reference category. MNA-SF—Mini Nutritional Assessment short-form; DVS—diet variety score; HR-QoL—health-related quality of life. Model 1—crude; model 2—adjusted by variables in the table (total MNA-SF scores and total DVS); model 3—adjusted by gender, age, diseases, and variables in the table (total MNA-SF scores and total DVS). Data are presented as odds ratio (OR) and confidence interval (CI). * Significance at *p*-value ≤ 0.05.

## Data Availability

Data are available upon reasonable request to the corresponding author.

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
