# Peer review of "Association of Nutritional Status and Diet Diversity with Skeletal Muscle Strength and Quality of Life among Older Arab Adults: A Cross-Sectional Study"

_nutrients, 2023, doi:10.3390/nu15204382_

Round 1

Reviewer 1 Report

The authors relate to nutritional status, nevertheless, blood albumin is not mentioned nor other strict parameters of nutrition. Similarly, the authors relate to robustness of muscle mass, nevertheless, there is no mentioning of either sarcopenia and frailty?

The manuscript should be revised relating to sarcopenia, frailty and more parameters of nutritional status or... explain their absence.

Author Response

Response to Reviewers

Manuscript ID: nutrients-2629236

Manuscript Title: “Association of Skeletal Muscle Strength with Nutritional Status and Quality of Life among Arab Older Adults: A Cross-Sectional Study”

We thank the reviewers for their careful examination of the manuscript and appreciate the useful suggestions to improve the quality of our paper. Our point-by-point response to the reviewers' comments is given below. Changes in the manuscript are tracked and marked in red color. Please note that the pages and line numbers mentioned in the reviewers’ comments refer to the original manuscript, whereas those in the authors’ reply refer to the revised manuscript.

Comments from the Reviewers:

Reviewer 1

  1. The authors relate to nutritional status, nevertheless, blood albumin is not mentioned nor other strict parameters of nutrition.

Response:

We agree with you regarding blood albumin as a good indicator of nutritional status. However, the study aimed to investigate whether the mini-nutritional assessment short form (MNA-SF), considered a well-validated, most commonly used tool to assess older adults' nutritional status, is associated with skeletal muscle strength and HR-QoL in older adults. Also, we mentioned this point in the study's limitation (We did not measure nutritional biomarkers in the blood).

  1. Similarly, the authors relate to robustness of muscle mass, nevertheless, there is no mentioning of either sarcopenia or frailty?

Response:

Thank you for the valuable comment. However, we decided not to write in detail on sarcopenia or frailty, as muscle strength alone cannot be considered a good indicator of sarcopenia or frailty in the absence of the measurement of muscle mass and function. In addition, we used a health-related quality of life (SF-36) questionnaire, which indicates physical and mental health, such as the ability to move and energy.

  1. The manuscript should be revised relating to sarcopenia, frailty, and more parameters of nutritional status or... explain their absence.

Response:

Same as with the response to your valuable comment number 2. We agree that muscle strength is related to sarcopenia and frailty, but it is not enough to indicate the presence or absence of these conditions. That is why we concentrated on handgrip strength as a measure of muscle strength status and reviewed some studies that highlighted the consequences of low muscle strength in older adults. With nutritional status concerns, we used the mini-nutritional assessment short form (MNA-SF), as it is a well-validated, easy-to-use tool to assess the nutritional status of older adults. Also, we used a health-related quality of life (SF-36), a validated questionnaire indicating participants' physical and mental health. Moreover, we mentioned this point in the study's limitation (We did not measure muscle mass).

Reviewer 2 Report

Comments: This study investigates the association of skeletal muscle strength with nutritional status and quality of life among Arab older adults. This study is interesting, complete and well-designed. But some minor questions need to be addressed before publication.

1. I noticed that the authors evaluated Health-Related Quality of Life (HR-QoL) using the Short Form-36 questionnaire. Is it a widespread method to evaluation? and also the correlation between this parameter and quality of life should be mentioned in the section of introduction.

2. In lines 25, “DVS was not associated with muscle strength and HR-QoL”. How to explain this result, as diet variety may refer to nutritional property of daily food intake.

Author Response

Response to Reviewers

Manuscript ID: nutrients-2629236

Manuscript Title: “Association of Skeletal Muscle Strength with Nutritional Status and Quality of Life among Arab Older Adults: A Cross-Sectional Study”

We thank the reviewers for their careful examination of the manuscript and appreciate the useful suggestions to improve the quality of our paper. Our point-by-point response to the reviewers' comments is given below. Changes in the manuscript are tracked and marked in red color. Please note that the pages and line numbers mentioned in the reviewers’ comments refer to the original manuscript, whereas those in the authors’ reply refer to the revised manuscript.

Comments from the Reviewers:

Reviewer 2

  1. Comments: This study investigates the association of skeletal muscle strength with nutritional status and quality of life among Arab older adults. This study is interesting, complete and well-designed. But some minor questions need to be addressed before publication.

Response: That is a perfect summarization, thanks.

  1. I noticed that the authors evaluated Health-Related Quality of Life (HR-QoL) using the Short Form-36 questionnaire. Is it a widespread method to evaluation? and also the correlation between this parameter and quality of life should be mentioned in the section of introduction.

Response: It is a widespread questionnaire used in various studies, and RAND validates it. In addition, the validity of the Short form-36 questionnaire to assess the quality of life in older adults was added to the introduction (in red color).

  1. In lines 25, “DVS was not associated with muscle strength and HR-QoL”. How to explain this result, as diet variety may refer to nutritional property of daily food intake.

Response: Thank you for your valuable notice. Table 6 shows that HR-QoL had a trend towards better scores with high DVS, particularly with total physical components: “DVS has a significant and weak correlation with total physical components (r=0.19) and total score of HR-QoL (r=0.16)”. Unfortunately, there is no significant correlation between diet variety and muscle strength, BMI, or CC. The explanation of non-significant association was added to the discussion part (in red color).

Reviewer 3 Report

This study attempted to investigate the association between nutritional status and diet diversity with skeletal muscle strength and health-related quality of life in older adults. Overall, I think this is an informative paper and should be useful in the relevant field. But prior to further consideration of this paper, the authors should revise it and clarify the issues in line with the following suggestions/comments. Line number/s have been mentioned for each comment as necessary.

Important issues:

1.      Introduction:  In the last paragraph under Introduction the authors cited a reference and mentioned (L62-63): “Taken together, further investigation into the associations between nutritional status, diet diversity, and key health outcomes in older adults is required”. However, from the prior paragraphs under Introduction, it is not clear what is lacking in the literature or what gaps in the existing body of knowledge this research aims to fill in, and why further investigations are required on such associations. The authors should justify the need for this study clearly and convincingly. The second paragraph under Introduction contains several repetitions of the same information.

2.   Methods: The authors should clearly describe and justify the cutoff values or criteria for categorization of HGS, KES and HR-QoL used in the logistic regression analyses (moderate and high versus low) as the important findings of this study are based on such classifications.

3.  Discussion: According to this reviewer, the discussion requires substantial revision as the authors did not discuss the results and their implications in an organized way and orderly manner, in line with the observed outcomes. Also, the reviewer would like to suggest the authors to emphasize on the results of logistic regression analyses of different variables and not to focus heavily on the significant results of correlation analyses that include the same variables. Also, please find below for some specific comments on Discussion.

4.      The whole manuscript should be revised for typos, English and scientific writing, and presented in an organized way without unnecessary repetition of texts. Such issues exist throughout the whole manuscript. It is not possible for this reviewer to point out and explain each and every of such issues related to this manuscript. However, some typical examples include: L12, L42-43 (clinical implications), L68 (A cross-sectional study included...), L73, L82, L83-84, L97-99 (these two sentences contain same information); L 108 (deleted), L172 (…chart in study…), L186-187 (malnutrition or malnutrition); L215 (knee extinction); L 228 (the sentences are wrongly expressed); L237-238 and L240-241 (repetition); L349 (dietary what); L412 (highly reliable what)ï¼› L416 (A study limitation can be enumerated) etc. and many more!

Some other comments:

1.      Title: From the text, it is understandable that the main objective was to explore the association of nutritional status and diet diversity with skeletal muscle strength and health-related quality of life in older adults. The current title does not reflect the content of this study.

2.      Abstract: Please express the conclusion in line with the observed findings. Also, the authors need to organize the text under Abstract appropriately. For example (L19), it might be better to mention the recruited study population earlier.

3.      L156: ‘between the variables’ is a vague expression.

4.      L159-160: The term ‘table’s variables’ seems not appropriate here. Instead, the authors can express as ‘variables in model 2’.

5.      L160: The P-value is one-sided or two-sided?

6.      Figure 1: It may be better to place the squares on the right side of arrows (information on exclusion) in a square placed between the main upper and lower squares as applicable [Vandenbroucke, Jan P et al. “Strengthening the Reporting of Observational Studies in Epidemiology (STROBE): explanation and elaboration.” Annals of internal medicine vol. 147,8 (2007): W163-94].

7.      Footnote for Table 1 (L191): The author cannot see any data presented as mean±sd.

8.      Table 3: Only in this table, the results of correlation analyses have been presented also for both males and females, and the purpose for it is not clear. It necessarily raises the question of presenting the findings of the research stratified by sex.

9.      L240-241 and Tables 6, 7: The authors should justify the need for categorizing DVS scores into 3 groups (tertiles). On the other hand, in Tables 8 and 9, the authors probably used the total DVS score to investigate its relation with other variables. If necessary, the information presented in Tables 6-7 (and Table 8) may be presented concisely using texts only as no significant observation could be revealed here.

10.  L266-282: Please write OR with corresponding 95% CI.

11.  Please express the titles of all tables and figures (and also footnotes) appropriately.

12.  Discussion:  a) 1st paragraph under discussion: The authors cited the prevalence rates or risks of malnutrition reported in other studies and commented in the last sentence of this paragraph as: “Therefore, malnutrition among older adults is a global issue that impacts economic development, aging, and education”. The link between the observed findings and relevant discussion with the last sentence is not understandable. b) 2nd paragraph under discussion: Here, at the beginning, the authors focused on the association of nutritional status with marital status and/or educational level (L318-322). But the subsequent sentences (L322-323) are not in line with the earlier mentioned words. c) 3rd and 4th paragraphs under Discussion (L329-355): Please rephrase the texts in an organized way without repetition of information. d) 5th paragraph under Discussion (L356-365): The reviewer probably missed, but could not find the results that showed significant gender differences in HGS, KES, CC, and BMI etc. as mentioned by the authors (L358).

The whole manuscript should be revised for typos, English and scientific writing.

Author Response

Response to Reviewers

Manuscript ID: nutrients-2629236

Manuscript Title: "Association of Skeletal Muscle Strength with Nutritional Status and Quality of Life among Arab Older Adults: A Cross-Sectional Study"

We thank the reviewers for carefully examining the manuscript and appreciate the valuable suggestions to improve the quality of our paper. Our point-by-point response to the reviewers' comments is given below. Changes in the manuscript are tracked and marked in red color. Please note that the pages and line numbers mentioned in the reviewers' comments refer to the original manuscript, whereas those in the authors' reply refer to the revised manuscript.

Comments from the Reviewers:

Reviewer 3

  1. This study attempted to investigate the association between nutritional status and diet diversity with skeletal muscle strength and health-related quality of life in older adults. Overall, I think this is an informative paper and should be useful in the relevant field. But prior to further consideration of this paper, the authors should revise it and clarify the issues in line with the following suggestions/comments. Line number/s have been mentioned for each comment as necessary.

Response: That is a perfect summarization, thanks. We will do our best to fix any problems.

------------------------------------------------------------------------------.

  1. Important issues:
  2. Introduction:In the last paragraph under Introduction, the authors cited a reference and mentioned (L62-63): "Taken together, further investigation into the associations between nutritional status, diet diversity, and key health outcomes in older adults is required". However, from the prior paragraphs under Introduction, it is not clear what is lacking in the literature, what gaps in the existing body of knowledge this research aims to fill in, and why further investigations are required on such associations. The authors should justify the need for this study clearly and convincingly. The second paragraph under the Introduction contains several repetitions of the same information.

Response:

Thank you for your valuable comment; we agree. The following sentence was added to highlight the gaps in the literature. "The association between nutritional status, using both the MNA-SF and diet diversity, and muscle strength and quality of life in older adults is scarce, and it has not been studied in the Kingdom of Saudi Arabia." Regarding "The second paragraph under the Introduction contains several repetitions of the same information,". The following sentence was deleted to avoid repetition. "Moreover, they affect functional mobility, leading to changes in nutritional requirements, malnutrition, and reduced muscle strength [6]. Meanwhile"

-----------------------------------------------------------------------------------

  1. Methods: The authors should clearly describe and justify the cutoff values or criteria for categorization of HGS, KES, and HR-QoL used in the logistic regression analyses (moderate and high versus low) as the important findings of this study are based on such classifications.

Response:

We agree with you; the explanation is added in the statistical analysis (in red).

Statistics

HGS

KES

Total_HRQOL

Percentiles

33 (low)

≤19.00

≤9.5000

≤59.86

66 (moderate)

>19..00  up to 23.00

>9.50  up to 11.82

>59.86  up to 82.00

100  igh)

 >23.00  up to 43.00

>11.82  up to 23.00

> 82.00 up to112.78

HGS, KES, and HR-QoL values were divided into three tertiles (high, moderate, and low), and then the high and moderate tertiles were combined as one group.

------------------------------------------------------------------------------.

  1. Discussion:According to this reviewer, the discussion requires substantial revision as the authors did not discuss the results and their implications in an organized way and orderly manner in line with the observed outcomes. Also, the reviewer would like to suggest the authors to emphasize on the results of logistic regression analyses of different variables and not to focus heavily on the significant results of correlation analyses that include the same variables. Also, please find below for some specific comments on Discussion.

Response: thanks for this comment. The critical difference between correlation and regression is that correlation measures the degree of a relationship between two independent variables (MNA-SF and HGS). In contrast, regression is how MNA-SF predicts, optimizes, or explains HGS. We modified the discussion to be more related to regression results. Focus on correlations because of our aim, which is the study of “the association”.

------------------------------------------------------------------------------.

  1. The whole manuscript should be revised for typos, English, and scientific writing and presented in an organized way without unnecessary repetition of texts. Such issues exist throughout the whole manuscript. This reviewer can't point out and explain each and every of such issues related to this manuscript. However, some typical examples include: L12, L42-43 (clinical implications), L68 (A cross-sectional study included...), L73, L82, L83-84, L97-99 (these two sentences contain the same information); L 108 (deleted), L172 (…chart in study…), L186-187 (malnutrition or malnutrition); L215 (knee extinction); L 228 (the sentences are wrongly expressed); L237-238 and L240-241 (repetition); L349 (dietary what); L412 (highly reliable what)ï¼›L416 (A study limitation can be enumerated) etc. and many more!

Response: Sorry for these oversights. All comments were fixed, and the manuscript was revised by a native English-speaking editor (MDPI editing services). The certificate is attached.

------------------------------------------------------------------------------.

  1. Some other comments:
  2. Title: From the text, it is understandable that the main objective was to explore the association of nutritional status and diet diversity with skeletal muscle strength and health-related quality of life in older adults. The current title does not reflect the content of this study.

Response:

Thanks, We agree with you. We changed the title to "Exploring the Association of Nutritional Status and Diet Diversity with Skeletal Muscle Strength and Health-Related Quality of Life among Arabs Older Adults."

------------------------------------------------------------------------------.

  1. Abstract: Please express the conclusion in line with the observed findings. Also, the authors need to organize the text under Abstract appropriately. For example (L19), it might be better to mention the recruited study population earlier.

Response: We agree with you. Done

------------------------------------------------------------------------------.

  1. L156: 'between the variables' is a vague expression.

Response:

Thank you. We agree with you, and the sentence was changed (in red color).

"Binary logistic regression was utilized to estimate the odds ratio and to assess the factors associated with nutritional status."

------------------------------------------------------------------------------.

  1. L159-160: The term 'table's variables' seems not appropriate here. Instead, the authors can express as 'variables in model 2'.

Response:

Thank you for your comments. ''Variables in model 2'' was written in the statistical analysis (in red color). 

------------------------------------------------------------------------------.

  1. L160: The P-value is one-sided or two-sided?

Response:

It is two-sided, and we add it to the statistical analysis (in red).

------------------------------------------------------------------------------.

  1. Figure 1: It may be better to place the squares on the right side of arrows (information on exclusion) in a square placed between the main upper and lower squares as applicable [Vandenbroucke, Jan P et al. "Strengthening the Reporting of Observational Studies in Epidemiology (STROBE): explanation and elaboration." Annals of internal medicine 147,8 (2007): W163-94].

Response:

Thank you. Done according to your valuable comment.

------------------------------------------------------------------------------.

  1. Footnote for Table 1 (L191): The author cannot see any data presented as mean±

Response:

Thank you. Sorry for this mistake. The mean±sd was deleted from the footnote. We used median and IQR for the age, and the data presented as frequencies for the other variables.

Footnote: "MNA-SF: mini nutritional assessment – short form; HTN: hypertension; CVD: cardiovascular disease; DM: diabetes mellitus. For the age, data was presented as median and IQR. Other variables were presented as frequencies and percentages. Significance at p-value < 0.05."

------------------------------------------------------------------------------.

  1. Table 3: Only in this table the results of correlation analyses have been presented for both males and females, and the purpose for it is not clear. It necessarily raises the question of presenting the findings of the research stratified by sex.

Response:

Thank you for your helpful comment. We agree that Table 3 needed modification to present the total sample consistently with other tables, and we deleted the male and female components.

------------------------------------------------------------------------------.

  1. L240-241 and Tables 6, 7: The authors should justify the need for categorizing DVS scores into 3 groups (tertiles). On the other hand, in Tables 8 and 9, the authors probably used the total DVS score to investigate its relation with other variables. If necessary, the information presented in Tables 6-7 (and Table 8) may be presented concisely using texts only as no significant observation could be revealed here.

Response:

Thank you for your valuable comments. Tables 6-8 were deleted, and results are presented using only text.

------------------------------------------------------------------------------.

  1. L266-282: Please write OR with the corresponding 95% CI.

Response:

Thank you for your comment. We have added OR and CI in the revised manuscript (in red color).

------------------------------------------------------------------------------.

  1. Please express the titles of all tables and figures (and also footnotes) appropriately.

Response: All were revised

------------------------------------------------------------------------------.

  1. Discussion:  a) 1st paragraph under discussion: The authors cited the prevalence rates or risks of malnutrition reported in other studies and commented in the last sentence of this paragraph as: "Therefore, malnutrition among older adults is a global issue that impacts economic development, aging, and education". The link between the observed findings and relevant discussion with the last sentence is not understandable.

Response:

Thank you for your comments. We changed the sentence to "Malnutrition among older adults is considered a global issue, and applying nutrition screening protocols becomes necessary to detect malnutrition at an earlier stage" (in red color).

------------------------------------------------------------------------------.

  1. b) 2nd paragraph under discussion: Here, at the beginning, the authors focused on the association of nutritional status with marital status and/or educational level (L318-322). But the subsequent sentences (L322-323) are not in line with the earlier mentioned words.

Response:

Thank you for your valuable comments. This was modified, now the 2nd paragraph based on the study's socio-demographic and clinical characteristics (red color in the revised manuscript), and it talks about clinical and socio-demographic characteristics, including marital status, educational level, and osteoporosis.

------------------------------------------------------------------------------.

  1. c) 3rd and 4th paragraphs under Discussion (L329-355): Please rephrase the texts in an organized way without repetition of information.

Response:

Thank you for your comments. We summarized and rephrased it without repetition.

------------------------------------------------------------------------------.

  1. d) 5th paragraph under Discussion (L356-365): The reviewer probably missed, but could not find the results that showed significant gender differences in HGS, KES, CC, and BMI etc. as mentioned by the authors (L358).

Response:

Thank you. We deleted significant gender differences from the discussion part, as the results were modified to present the total sample based on the response to your respective comment number 13.

------------------------------------------------------------------------------.

  1. The whole manuscript should be revised for typos, English and scientific writing.

Response: the manuscript was revised by a native English-speaking editor (MDPI editing services). The certificate is attached.

------------------------------------------------------------------------------.
